# Babies, bugs and brains: How the early microbiome associates with infant brain and behavior development

Sebastian Hunter[1], Erica Flaten[2], Charisse Petersen[3,4], Judit Gervain[5,6,7], Janet F. Werker[8], Laurel J. Trainor[2,9,10], Brett B. Finlay[1,11,12]*

1 Michael Smith Laboratories, University of British Columbia, Vancouver, BC, Canada, 2 Department of Psychology, Neuroscience and Behaviour, McMaster University, Hamilton, Ontario, Canada, 3 Department of Pediatrics, BC Children's Hospital, University of British Columbia, Vancouver, BC, Canada, 4 British Columbia Children's Hospital, Vancouver, BC, Canada, 5 University of Padua, Department of Developmental and Social Psychology, Padua, Italy, 6 University of Padua, Padova Neuroscience Center, Padua, Italy, 7 Université Paris Cité & CNRS, Integrative Neuroscience and Cognition Center, Paris, France, 8 Department of Psychology, University of British Columbia, Vancouver, BC, Canada, 9 McMaster Institute for Music and the Mind, McMaster University, Hamilton, Ontario, Canada, 10 Rotman Research Institute, Baycrest Hospital, Toronto, Ontario, Canada, 11 Department of Microbiology and Immunology, University of British Columbia, Vancouver, BC, Canada, 12 Department of Biochemistry and Molecular Biology, University of British Columbia, Vancouver, BC, Canada

* bfinlay@msl.ubc.ca

**Data Availability Statement:** All files are available at Figshare (DOIs 10.6084/m9.figshare.22728368, 10.6084/m9.figshare.22728371, 10.6084/m9.

## Abstract

Growing evidence is demonstrating the connection between the microbiota gut-brain axis and neurodevelopment. Microbiota colonization occurs before the maturation of many neural systems and is linked to brain health. Because of this it has been hypothesized that the early microbiome interactions along the gut-brain axis evolved to promote advanced cognitive functions and behaviors. Here, we performed a pilot study with a multidisciplinary approach to test if the microbiota composition of infants is associated with measures of early cognitive development, in particular neural rhythm tracking; language (forward speech) versus non-language (backwards speech) discrimination; and social joint attention. Fecal samples were collected from 56 infants between four and six months of age and sequenced by shotgun metagenomic sequencing. Of these, 44 performed the behavioral Point and Gaze test to measure joint attention. Infants were tested on either language discrimination using functional near-infrared spectroscopy (fNIRS; 25 infants had usable data) or neural rhythm tracking using electroencephalogram (EEG; 15 had usable data). Infants who succeeded at the Point and Gaze test tended to have increased Actinobacteria and reduced Firmicutes at the phylum level; and an increase in *Bifidobacterium* and *Eggerthella* along with a reduction in *Hungatella* and *Streptococcus* at the genus level. Measurements of neural rhythm tracking associated negatively to the abundance of *Bifidobacterium* and positively to the abundance of *Clostridium* and *Enterococcus* for the bacterial abundances, and associated positively to metabolic pathways that can influence neurodevelopment, including branched chain amino acid biosynthesis and pentose phosphate pathways. No associations were found for the fNIRS language discrimination measurements. Although the tests were underpowered due to the small pilot sample sizes, potential associations were identified between

figshare.22728374,10.6084/m9.figshare.
22728377.).

**Funding:** This research was funded by grants to
JFW, LJT and BBF from the Canadian Institute for
Advanced Research (CIFAR, https://cifar.ca/) (FL-
000981-CF, FL-000982-CF, & FL-000983-CF).
Work in B.B.F.'s lab is also supported by a
Canadian Institute for Health Research (CIHR)
Foundation Grant, in J.F.W.'s lab by a Natural
Sciences and Engineering Research Council of
Canada (NSERC) Discovery Grant and a Social
Sciences and Humanities Research Council of
Canada (SSHRC) Insight Grant, and in L.J.T.'s lab
by grants from CIHR, NSERC and SSHRC. The
funders had no role in study design, data collection
and analysis, decision to publish, or preparation of
the manuscript.

**Competing interests:** The authors have declared
that no competing interests exist.

the microbiome and measurements of early cognitive development that are worth exploring
further.

## Introduction

Infant brain and behavior development is a dynamic and protracted process that starts a few
weeks after conception with the formation of the central nervous system (CNS) through pro-
cesses such as neurulation, neuron migration, and synaptogenesis, among others [1, 2]. Before
birth all the nervous system structures are formed, although with various degrees of matura-
tion, and development continues postnatally until adulthood when it reaches maturity [1, 2].
The first two years after birth are characterized by rapid changes in the nervous system, such
as a significant increase in synaptic growth and subsequent pruning, the proliferation of glial
cells (astrocytes, microglia and oligodendrocytes) and increasing myelinization, all which
result in the expansion of the infant´s brain to more than double that of its birth size [2–5].
Coupled with structural growth, functional networks also develop at a great speed during the
first years after birth, including auditory, visual, and sensorimotor networks, enabling the
emergence of more complex skills [2, 5–7].

Infant neurodevelopment is shaped by precise temporal control of brain circuit formation
and signaling pathways of hormones, neurotransmitters, and immune cells [2, 3, 8, 9]. For this
intricate regulation the CNS interacts with other systems, such as the endocrine, immune, and
enteric nervous system (ENS), with the latter being of special interest due to its interconnectiv-
ity via the gut-brain axis (GBA) [8, 10]. The interplay between the brain and the GI tract is cru-
cial as the GBA can modify and regulate cognitive functions and mood, and nutritional
compounds transported by the gut can have an effect on brain development [10, 11]. In the
last couple decades, there has been a great interest in determining the connection between the
gut microbiota diversity and metabolic capacity to brain health and development.

The microbiota gut-brain axis is an intricate communication network between the gut
microbiome, the gastrointestinal tract, and the nervous system. The microbiota has been
shown to influence and predict brain health in adulthood, and its absence in germfree mice
results in the development of abnormal brain functions [12, 13]. Despite the current knowl-
edge it is unclear how the microbiota acts within the gut-brain axis to build a healthy brain.
Experiences in early life are the most impactful for brain development, especially during
infancy when it is most acutely sensitive to opportunities and insults [14, 15]. Microbiome col-
onization begins prior to the maturation of many neural systems and is linked to later brain
health, suggesting commensal microbes likely play a role in brain maturation [16]. It has been
hypothesized that early microbiota interactions along the gut-brain axis evolved to promote,
and therefore may predict, advanced brain and behavioral development.

Studies have reported how the microbiota can impact the brain development of infants,
either prenatally via the mother's microbiome, or postnatally, where factors such as delivery
method, breastfeeding and antibiotic usage can alter the healthy infant microbiome composi-
tion, resulting in neurodevelopmental changes [5, 11, 17]. Perturbation to the mother's micro-
biome during the prenatal period can affect fetal mental and physiological development,
mainly through metabolic dysregulations [9, 10]. Studies in mice have shown that the adminis-
tration of antibiotics during pregnancy can result in increased anxiety behaviors and reduced
sociability in their offspring [18, 19]. In a mouse model of prenatal stress, an association was
identified between changes in the vaginal microbiome and metabolic processes, which are cru-
cial for proper neurodevelopment [10, 20]. Postpartum, the establishment of a healthy micro-
biome is important as an unbalanced colonization can lead to metabolic changes and promote

opportunist pathogens. These unhealthy microbiomes have been linked to increased risk of conditions such as asthma, coeliac disease, necrotizing enterocolitis, among others [21]. Although what constitutes a healthy microbiome during development hasn't been defined yet due to its complexity, certain tendencies have been observed for the natural progression of the microbiome from birth to adulthood [22]. After birth infants are predominantly colonized by bacteria from the mother's microbiome, mostly *Lactobacillus* and *Bifidobacterium*. After weaning, the type of diet shifts the microbial composition (increased Firmicutes in carbohydrate diets; increased Bacteroidetes in animal protein diets). At one year of age the microbiome is characterized by high levels of *Akkermansia*, *Veillonela*, *Bacteroides* and *Clostridium* [23]. Microbial diversity increases steadily into adulthood where it stabilizes [23]. The microbiota can produce metabolites that can influence the nervous system, including neurotransmitters, hormones, and short chain fatty acids (SCFA) [10, 17, 24, 25]. The gut microbiota is capable of synthesizing dopamine, norepinephrine, gamma-aminobutyric acid (GABA) and serotonin, all of which can modulate the CNS, but it is unclear if these compounds can cross the blood brain barrier (BBB) and have a direct effect on the brain; nevertheless these compounds can have an impact on the local gut area, such as by reducing proinflammatory cytokines and regulating gut motility, among others [9]. Other metabolites synthesized by the microbiota are the SCFAs, metabolites produced by the fermentation of dietary fibers [10, 26, 27]. Not only do SCFAs modulate the gut by maintaining the intestinal barrier integrity, but they can also affect brain development through the modification of the BBB permeability, via microglia activation and neuroinflammation regulation [10, 26, 27].

Research is beginning to show how the microbiome can influence neurodevelopment during infancy, an important and dynamic period of brain growth whose characteristics can predict risk or resilience to neuropsychiatric disorders, either in childhood or later adulthood [28, 29]. In certain developmental disorders both dysregulations in the microbiome and impairments in cognitive functions and behaviors are present, although it is unclear how they are related or if they are in fact independent from one another [28, 30]. For example, in autism spectrum disorder, impairments in rhythmic processing and timing have been described, as well as an altered gut microbial composition [28, 30–32]. Previous studies have found associations between differential bacterial composition and cognitive performance based on the Mullen Scales of Early Learning (MSEL), particularly differences in expressive language, receptive language, and visual reception scales [28]. Although the MSEL is a standardized test to assess infant development, it does not include neural measures, which can provide deeper insight into the relationship between brain development and gut microbiota [33, 34]. In the present study we include neural measures and concentrate on three early-emerging abilities. Neural rhythm tracking is important for organizing information across time and it affects perception, social communication, language, and cognition [35–38]. Further, rhythm tracking deficits have been documented in developmental disorders [31, 39]. Discriminating language from non-language is an important and early emerging precursor to language acquisition. Joint attention is a social communicative skill that affects infants' ability to learn from others, thereby affecting early word learning and cognition in general. These are ideal systems to quantify neural development because of their well-documented developmental trajectories and neural underpinnings [31, 40–42].

Although several connections between the microbiome and brain development have been described in animal models and human cohorts, few studies have looked at how changes in the early infant microbiome correlate to differences in emerging cognitive capacities and behavioral development. Studying how the structure of the microbiome and the development of brain systems are associated can provide new insight into the interconnectivity between the brain and the gut. We performed a pilot study with a multi-disciplinary approach to quantify

the development of brain systems underlying perceptual and communicative development, as well as the composition of the microbiota in infancy, with a focus on microbial diversity and functional potential of the gut microbiome.

## Methodology

### Sample collection

Fecal samples from 56 infants were collected between the University of British Columbia (UBC) and McMaster University (McMaster) between 4 and 6 months of age ($N$ = 25 female, M age = 4.92 months). Samples were collected between February 15, 2019, and February 24, 2020. Two complementary brain imagining techniques were used in two separate groups of infants, one at each university study center, to quantify early brain development in the context of auditory cognitive tasks (language and music processing). From the 31 infants at UBC, 25 participated and produced usable data on the language processing task using functional near-infrared spectroscopy (fNIRS) in the laboratory of Janet Werker ($N$ = 9 female, M age = 4.38 months). From the 25 infants at McMaster, 15 participated and produced usable data on the musical rhythm tracking task using electroencephalogram (EEG) measurements at the laboratory of Laurel Trainor ($N$ = 6 female, M age = 5.56 months). Metadata was also collected about their age, sex, number of siblings, mode of delivery, use of antibiotics by the infants, introduction of solid foods to the infants and exposure to pets. For the selection criteria, infants were recruited using Research Databases at each university in 2019. Infants had to be full-term (37.5 weeks +) with normal hearing (assessed by universal newborn hearing screening in Ontario and British Columbia).

### Ethics statement

Informed written consent was obtained from the parent/legal guardian of each subject. The study was approved by the McMaster Research Ethics Board (MREB #: 0411), the UBC Behavioral Research Ethics Board (#: H18-00840) and UBC Clinical Research Ethics Board (PAA #: H18-02437).

### Functional near-infrared spectroscopy (fNIRS)

Functional near-infrared spectroscopy (fNIRS) is a brain imaging technique used here to measure the cortical hemodynamic response of infants while being exposed to forward and backwards speech in their maternal language [43]. Reverse speech preserves some of the properties of the language (e.g., some phonemes) but not all, and notably not prosody [44]. Importantly, it is an ideal control, as it has the same overall physical properties as forward speech, but all temporal information is compromised. Backwards speech is thus not processed as a linguistic signal [45]. Specifically, temporal brain areas responsible for speech perception have been previously found to show a greater hemodynamic response to forward speech as compared to backwards speech in the native language [46], indicating maturation of the cortical language system. The measurements were taken at UBC using a 42-channel (16 sources and 16 detectors) NIR Scout fNIRS system by NIRx Medical Technologies LLC with 760 and 850 nm wavelengths between February 28, 2019, and September 20, 2019.

The obtained NIRS data was pre-processed using a standard pipeline described in Gemignani & Gervain [47; pipeline A]. The data was then statistically analyzed using cluster-based permutation tests over oxygenated and deoxygenated hemoglobin to identify clusters of channels and time windows in which the forward and backward speech conditions significantly differed or in which one of the conditions significantly differed from a zero baseline, as is common in the infant NIRS literature [48].

The cluster-based permutation analyses yielded no significant differences between the forward and backward speech conditions. Importantly, however, both evoked distinct hemodynamic responses that were significantly greater than the baseline. Specifically, activation to backward speech was greater than the baseline in a cluster comprising channels 9, 11 and 12 (Cluster 1) in the left hemisphere (LH) and in a cluster comprising channels 22 and 24 (Cluster 2) in the right hemisphere (RH). The forward condition significantly activated the cluster of channels 1 and 10 (Cluster 3) in the LH and of channels 17, 22 and 24 (Cluster 4) in the RH. The mean oxyhemoglobin responses from each of these clusters over the significant time windows were used as a measurement to be associated with the microbiome data.

## Electroencephalogram (EEG)

Electroencephalography (EEG) is a brain imaging technique used here to measure the electrical activity of infant brains during presentation of a rhythmic stimulus [49]. To organize incoming music or speech, listeners tend to apply hierarchical structures to group or divide sound events. In music, we can perceptually group musical beats to create meter, and this process can be measured in brain activity measured by EEG. For example, when presented with an ambiguous 6-beat rhythm, adults instructed to imagine the rhythm in groups of 2 beats (duple meter, like a march) or 3 beats (triple meter, like a waltz) show stronger neural tracking at the frequency for that meter, as measured by energy in their steady state evoked potentials (SSEPs) [50]. Infants who passively listened to the same stimulus showed equivalent neural tracking for the duple, triple, and beat frequencies [51]. Further, when grouping a rhythm with a certain metrical structure, infants and adults show larger mismatch responses (MMR; an event-related potential elicited by violation of expectation [52, 53]) to changes that occur on strong, compared to weak beat positions [54–56]. However, the extent to which infants can maintain a metrical structure on an ambiguous rhythm by means of internal, or top-down, processes is less understood, and was the main research question for the EEG portion of the current study [40].

A detailed description of the stimuli, protocol, and findings are presented in Flaten et al., 2022 [40]. EEG data were captured by a child-friendly 124-channel HydroCel GSN net with an Electrical Geodesic NetAmps 200 amplifier and recorded with the Electrical Geodesic Net Station software (v.5.4.2) at Trainor's Lab between September 5, 2019, and March 3rd, 2020 [40]. Infants were presented with a repeating, ambiguous 6-beat auditory rhythm that could be perceived either in duple (three groups of 2 beats), or triple (two groups of 3 beats) meter. Infants were intermittently primed to hear the rhythm in either duple or triple meter, by inserting loudness accents on every second, or every third beat, respectively. To elicit MMRs, the ambiguous (unaccented) stimulus contained occasional pitch deviants on beat 4 (strong beat for triple meter; weak for duple meter) or beat 5 (strong beat for duple; weak for triple), with the expectation that MMR would be larger for pitch changes on strong, compared to weak beats, according to their priming [40, 57, 58]. To examine how infants tracked the rhythm after priming, frequency-tagging was used to measure the energy of infant's SSEPs at the rhythmic frequencies in the ambiguous stimulus: the basic beat, and duple, and triple meters, expecting that SSEP energies would be higher at the primed, vs. unprimed frequencies [50, 51]. Electrodes over the frontal left and frontal right sites were used for all analyses. Infants' MMR for beat 4 and 5, and their SSEPs at the beat, duple, and triple frequencies were compared between the duple and triple groups using mixed ANOVA methods. It was found that infants primed to hear the rhythm in duple showed larger MMRs for beat 5, compared to those primed to hear the rhythm in triple, who showed larger response for beat 4, suggesting top-down effects on meter perception inducted by the priming [40]. However, top-down effects were not found in the SSEPs, as groups did not differ across the rhythmic frequencies according to priming [40]. The list of EEG measurements can be found in S1 Table.

## Point and gaze following tests

The Point and Gaze test was performed on infants as a behavioral measure of joint attention in a social context to complement the neuroimaging techniques [59]. In contrast to the other measurements and tests, the Point and Gaze test was performed on 44 participating children across both sites (31 infants at UBC and 13 infants at McMaster). For the task the researcher sat behind a table across from the infant, who was sitting on their parent's lap (parents wore covered glasses so they could not affect their infant's attention). After the researcher got the infant's attention, they placed one of four rolling toys on the table and slowly rolled it to the other side to draw attention to the toy. A second toy was then rolled to the other side of the table to draw the infant's attention to that toy. Then, the researcher captured the infant's attention by smiling, waving, or calling them by name until the infant made eye contact. The experimenter then silently pointed and gazed at one of the two toys for seven seconds, lowered their head for five seconds, and looked up at the infant again, indicating that the trial was over. The process was repeated five times. The toys were then removed from the table and the process was repeated with the remaining two toys for the gaze test (the researcher only gazed at the toy for seven seconds). Again, this was repeated five times. The order and placement of the toys, as well as which toy the investigator gazed/pointed at were randomized across trials and infants.

The task was video-recorded and later analyzed to determine if the infant followed the experimenter's pointing and gazing. Data coding, using Datavue, followed three steps: First, the onset of either a gaze, or point, from the experimenter–as indicated by the first frame wherein they began to turn their head, or raise their hand in a point, was marked. Second the end of the observation interval was marked by the onset of the research assistant looking down. Third, a second research assistant then went through the marked files, while only observing video of the infant as to remain blind to the correct direction, to code whether the infant looked to the left or the right following at any point during the observation interval. Only the first look from the infant during this interval was recorded. The L and R looks were then entered on a trial by trial basis, as correct or incorrect for the direction the experimenter was pointing, into the spread sheet for analysis. Each of the five trials were categorized as unsuccessful if the infant was paying attention but failed the task, successful if the infant followed the gazing/pointing of the experimenter, or invalid if the infant was not paying attention. The tests were considered successful if 66% or more of the completed trial were successful.

Previously, point- and gaze-following behaviors at young ages have been used to predict later vocabulary growth; participants with higher point or gaze following behaviors have been found to show greater increases in vocabulary size by age 2 years [60]. Because of its association to later language development, the Point and Gaze test was used as another metric to measure brain development that could be compared across all the participants at both testing sites.

## Stool collection

Stool samples were collected at home using the OMNIgene•GUT | OMR-200 kits from DNA genotek following manufacturing instructions. Tubes containing stabilized stool samples were then shipped at room temperature to UBC in prepaid envelopes. All samples were stored at -70˚C until downstream processing.

## DNA isolation

Stool DNA was isolated using QIAamp PowerFecal DNA Kit (QIAGEN Inc., Toronto, ON) according to the manufacturer's instructions. Picogreen quantification was conducted to

ensure that samples had minimum DNA concentrations of 1ng/μl. Samples were then shipped to Integrated Microbiome Resources (IMR; Dalhousie University, Halifax, NS) for sequencing.

Microbial genomes and community metagenomes are prepared using the Illumina Nextera Flex kit for MiSeq+NextSeq. Samples were enzymatically 'sheared' and tagged with adaptors, PCR amplified while adding barcodes, purified using columns or beads, normalized using Illumina beads or manually, then pooled for loading onto the Illumina Miseq.

## Microbiome analysis

Processing of the shotgun metagenomic reads was done under the Metagenomics standard operating procedure guide from the Langille Lab and the HMP Unified Metabolic Analysis Network (HUMAnN3) pipeline [61, 62]. After preprocessing, the dataset included 209 bacterial species as well as gene family and pathway feature to provide an in-depth microbiome functional analysis.

All statistical analysis and visualization were done in R [63]. Alpha diversity analysis was performed using the *vegan* package [64], calculating richness, Shannon index and Simpson index on the samples based on the collection center or the success in infants in the Point and Gaze test. The normality test by Shapiro-Wilks indicated normal distribution for the values in each category. A two-tailed t-test was used to compare the groups. Principal component analysis was performed and analysis with the packages *prcomp* (from the *stats* package), *factoextra*, and *ggplot2* [63, 65, 66]. PERMANOVA analysis was used to analyze the variance of the distance matrix, using the function *adonis* from the package *vegan* [64]. The PERMANOVA analysis was performed on the success on at least one of point and gaze subtests, with 999 permutations, using Euclidean distances to calculate the pairwise distances.

Relative abundance analysis was done at the phylum and genus taxonomic levels. Normality tests by Shapiro-Wilks indicated a non-normal distribution for the values. Microbiome compositional data was CLR transformed before differential analysis was performed. A general linear model was used to compare the microbial abundances, correcting for infant's age as a fixed effect and collection site as a random effect in the model using MaAsLin2 [67]. P-values were adjusted using FDR. Association analysis between the observational and quantitative outputs of the neural and behavioral analyses and the microbiota were performed to identify possible interactions between the microbiome and brain development. Multivariable associations between the microbiome meta-omics (microbe and pathways abundances) and the EEG/fNIRS measurements were done using MaAsLin2 [67]. The association models included covariates such as site (random effects) and infant age (fixed effects). Significant associations had to pass a q-value (p-value adjusted for multiple test comparisons) threshold of 0.05 and have a minimum prevalence of 10% across all samples.

## Results

### Associations between the microbiome and point- and gaze-following behavior

Infants were tested for their understanding of non-verbal joint attention communication using the Point and Gaze test. The test served as a metric of social skills that underlie and predict future language and cognitive development. It was measured in most participants across both sites, making it possible to directly compare infant development at the two sites. First an odds ratio analysis (OR) was performed between the outcome of the Point and Gaze subtests and the metadata from the infants to measure the association between them [68]. OR values were around one for most variables, except between the success in the gaze subtest and the

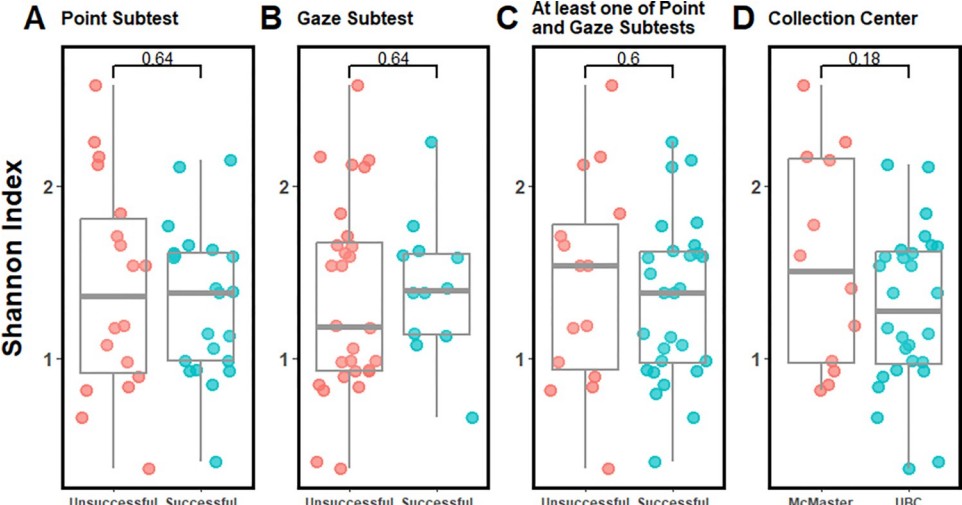

**Fig 1. Shannon diversity index and the point and gaze test of joint attention.** Microbial diversity analysis measured with Shannon index. Comparisons between infants who were unsuccessful versus successful on the point subtest (A), gaze subtest (B), or at least one of the point and gaze subtests (C), as well as between the collection site (UBC and McMaster) (D). Two-tailed t-tests were performed between the conditions. No statistical differences were observed between the groups.

mode of delivery of the children (vaginal versus cesarian section). For the point subtest none of the variables had higher or lower OR values than expected by chance (S1 Fig).

Alpha diversity was then analyzed on the microbiome data obtained from the participating infants. Three alpha diversity indexes were measured and compared between the participants according to their success on the point and gaze subtest, these were: richness, Shannon index and Simpson index. These distinct indexes measure different aspects of the community structure within the samples [69–71]. Fig 1 illustrates the difference in Shannon index between samples based on either their success in the point and gaze subtests or based on the sample collection site. No statistical differences were observed in the Shannon index between any of the comparisons ($p > 0.05$). Analyzing the other alpha diversity metrics, we found a statistical difference between samples collected from UBC and McMaster, with samples from UBC having less richness (S2 Fig, $p = 0.032$). No other differences were observed between the groups (S2 and S3 Figs). When analyzing the alpha diversity on the pathway abundance data no differences were observed between the analyzed groups (S4 Fig).

A Principal Components Analysis (PCA) was done with the microbiome abundance data, which was Center Log Ratio transformed (CLR) for a proper analysis of compositional data [72]. The first component explained around 17% of the variation between the samples, while the second highest component explained around 12% of the variance (Fig 2A). Performing a PERMANOVA analysis on the success on at least one of point and gaze subtests showed no statistical difference between children (Fig 2A). From the PCA analysis we selected the top contributing variables (bacteria species) that explained the variance. As shown in Fig 2B, we observe that the top five contributing bacteria to the principal components were *Bifidobacterium breve*, *Bifidobacterium longum*, *Clostridium neonatale*, *Bifidobacterium bifidum* and *Klebsiella pneumonia*.

Microbiome diversity differences at the phylum and genus levels were evaluated, as they provide key information related to the microbiota composition and their enterotype, as well as being the most described phylogenetic levels in the literature [23]. We compared the relative abundance of several taxa between children with successful or unsuccessful results in the Point

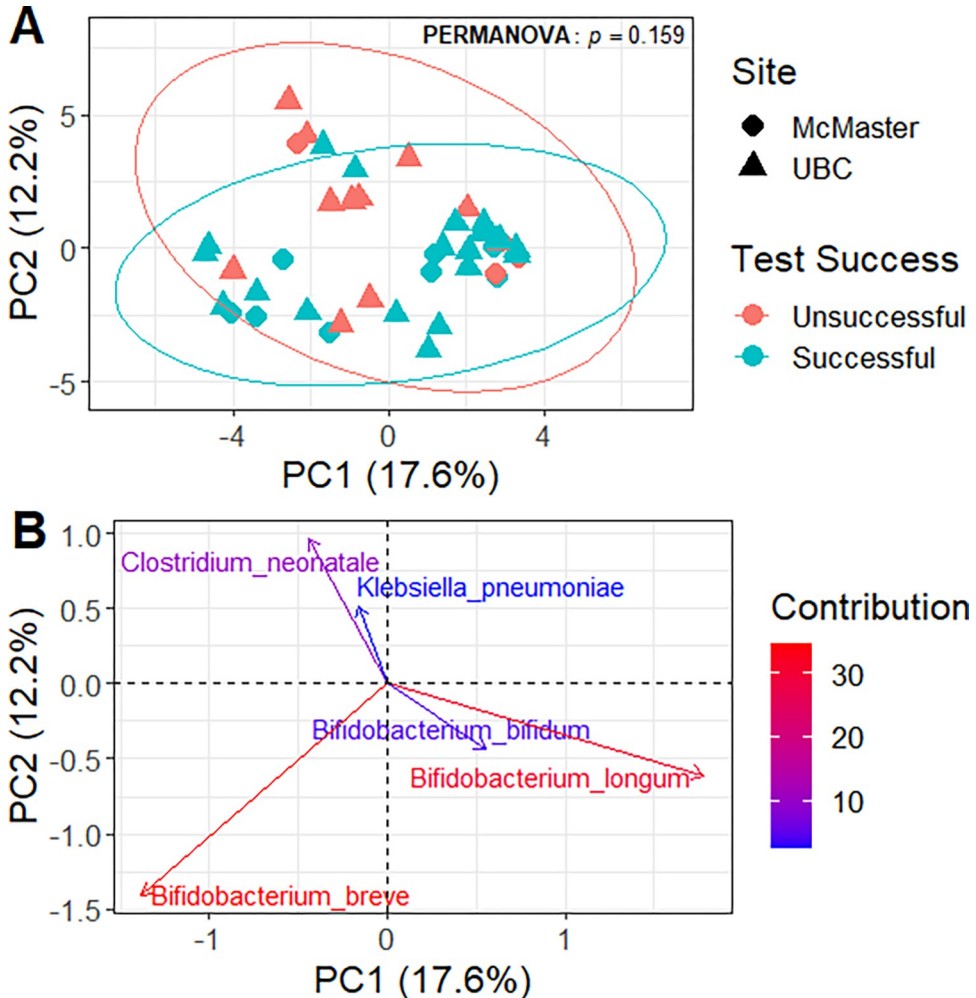

**Fig 2. Principle component analysis.** (A) PCA analysis on microbiome abundance data. Data was CLR transformed before the analysis. Success at least one of the point and gaze subtest tests are indicated in red and failure on both in green (95% confidence ellipse assuming a multivariate normal distribution) and shape indicates the sample site. (B) PCA analysis on the microbiome abundance data visualizing the top five contributing bacterial species (percentage %). Contribution is calculated as the cos2 of the variable divided the total cos2 of a given principal component: (var. cos2*100)/(total cos2 of the component).

and Gaze task. The relative abundance values were aggregated at the genus or phylum level and then normalized using a centered log ratio transformation (CLR). Comparisons were adjusted from the effect of age of the infants at the time of fecal sample collection and the site where the infants were tested (UBC or McMaster). At the phylum level we observed that children who were successful on at least one subtest of the Point and Gaze task had a higher relative abundance of Actinobacteria and a lower relative abundance of Firmicutes compared to infants who failed both subtests (Fig 3A, *p-value*); no differences were observed in the Bacteroidetes and Proteobacteria phylum. After FDR correction for multiple comparisons, none of the observed differences were statistically significant (Fig 3A, *q-value*). At the genus level, 31 bacteria passed the 10% prevalence cut-off, from which we observed six groups that were statistically significant before FDR correction. Two of the six genera are rare bacterial groups with a mean relative abundance under 0.5%; these genera were *Gordonibacter* and *Megasphaera*. The other genera were *Bifidobacterium* and *Eggerthella*, which had higher relative

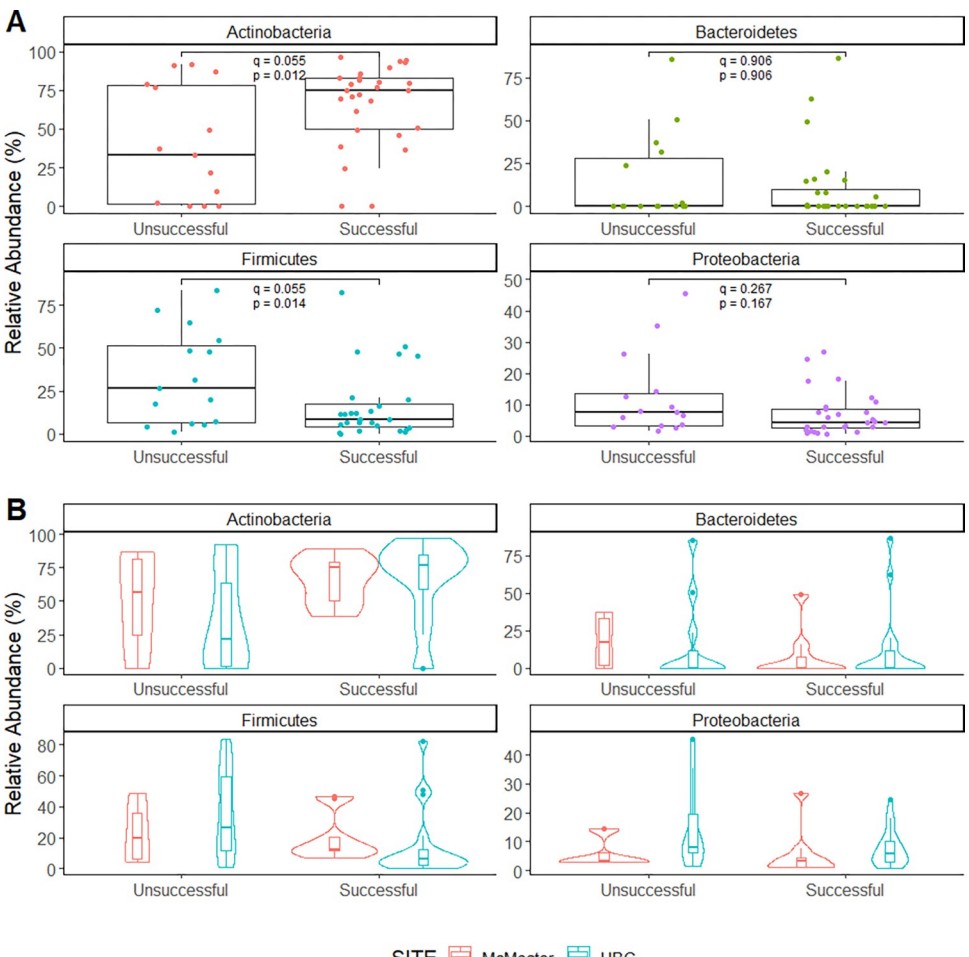

**Fig 3. Bacteria relative abundance.** Bacteria relative abundance at the phylum level. A) Difference in bacteria relative abundance by success on at least one of the point and gaze subtests versus not successful on both B) Violin plot of the relative abundance difference between collection site (McMaster and UBC) and joint attention test results.

abundance in participants who were successful in at least one subtest of the Point and Gaze test; in comparison, participants who failed both subtests had higher relative abundance of *Hungatella* and *Streptococcus* (S5A Fig). After multiple comparisons correction, none of these differences remained statistically significant (S5A Fig, *q-value*).

Visualizing the differences in the infant's microbiome composition by their sample site and their results on the Point and Gaze task, an interaction effect was observed between the variables affecting the microbial distribution and site (UBC, McMaster). At the phylum level, similar microbial distributions were observed between the two collection sites for the four analyzed phyla in infants with successful results on at least one of the point and gaze subtest (Fig 3B). However, for the infants who failed both subtests, we observe differences between sites. Actinobacteria and Firmicutes had similar distributions (higher mean value of Actinobacteria in samples from McMaster compared to those from UBC, while the phyla Bacteroidetes and Proteobacteria had different arrangements (specifically, for the Bacteroidetes phylum, samples from McMaster had an even distribution, while samples from UBC were highly concentrated at lower abundance levels with few samples at higher levels; for the Proteobacteria phylum the McMaster samples were centered at a relative abundance under 5% with few samples over 10%

relative abundance, whereas the samples from UBC had an even spread with a mean value around 10%, and with some samples reaching values over 40%) (Fig 3B). At the genus level we observed different patterns by Point and Gaze success and site. Some genera have different distributions based on the Point and Gaze performance, with no difference in site, such as *Bifidobacterium*, *Clostridium*, *Eggerthella* and *Klebsiella* (S5B Fig). Other genera differ mainly by the site of the sample, as in *Escherichia*, where UBC samples have higher abundances, changing the clustering (S5B Fig). Lastly, there were some interactions between variables. Specifically, *Hungatella* and *Streptococcus*, was common only in UBC infants who failed on both point and gaze subtests; and *Bacteroides* and *Veillonella* in the McMaster participants who failed on both point and gaze subtests had a different distribution compared to the other groups (S5B Fig).

## Associations between the microbiome and brain imaging (EEG and fNIRS) measurements for rhythm and language processing

Multivariable association analyses were then performed between the measurements from the two complementary brain imaging techniques, EEG and fNIRS, and the metagenomic microbiome data, corrected for FDR. Two models were used in the multivariable association tests, one performing a linear model analysis between the metagenome data and the brain imaging measurements, with the data normalized by CLR; the second model mirroring the first but including age of the infants as a fixed effect, to account for the age differences between the participants. Fig 4A shows the bacteria abundance values at the genus level for the first model. Associations were only found between bacteria genera and EEG measurements. Of the eleven EEG variables studied, only one had statistically significant associations, which was the SSEP amplitudes at the beat frequency in the rhythmic stimulus in the frontal left sites (BEAT_FL). BEAT_FL had a positive association with the genus *Clostridium* and *Enterococcus*, and a negative association with the genus *Bifidobacterium*, as observed in Fig 4A and S6 Fig. To test if the observed signal was only present in the frontal left hemisphere a new variable was calculated from the average of all the electrodes across the scalp, which we called BEAT_ALL. After incorporating this variable in the analysis, again only significant associations between BEAT_FL and the bacteria genera were found. The same analysis was performed at the species level, and again BEAT_FL was the only variable with statistically significant associations (S7 Fig). The only significant association found was between BEAT_FL and *Enterococcus faecalis*, with a coefficient of 0.65 (S8A Fig). In the second statistical model (corrected for age), BEAT_FL was again the only variable to have significant associations, limited to a positive association to *Enterococcus* and *Enterococcus faecalis* at the genus and species levels, respectively (S2 Table).

Association analysis between the pathway abundances predicted from the metagenomic samples and the fNIRS/EEG measurements was performed using the same statistical models described above for the bacterial associations. Multivariable association analysis was performed on the unstratified, and a subset of the stratified, metabolic pathways defined by MetaCyc. For the unstratified metabolic pathways only one variable from the EEG/fNIRS measurements had significant associations in the first statistical model. BEAT_FL, had 57 positively associated pathways (19 related to biosynthesis, 33 related to degradation, utilization, and assimilation, and 5 related to generation of precursor metabolites and energy) and 17 negatively associated pathways, all from the biosynthesis super class (S9A Fig). BEAT_FL had positive associations with amino acids, carbohydrates, lipids, nucleotides, and energy metabolism pathways, while it had negative associations only to amino acid, carbohydrate, and nucleotide metabolism pathways (S9B Fig). At the class level BEAT_FL had more positive associations to pathways related to degradation and salvage, while many of the biosynthesis pathways were negatively associated (S9C Fig). Some examples are the positive association between BEAT_FL

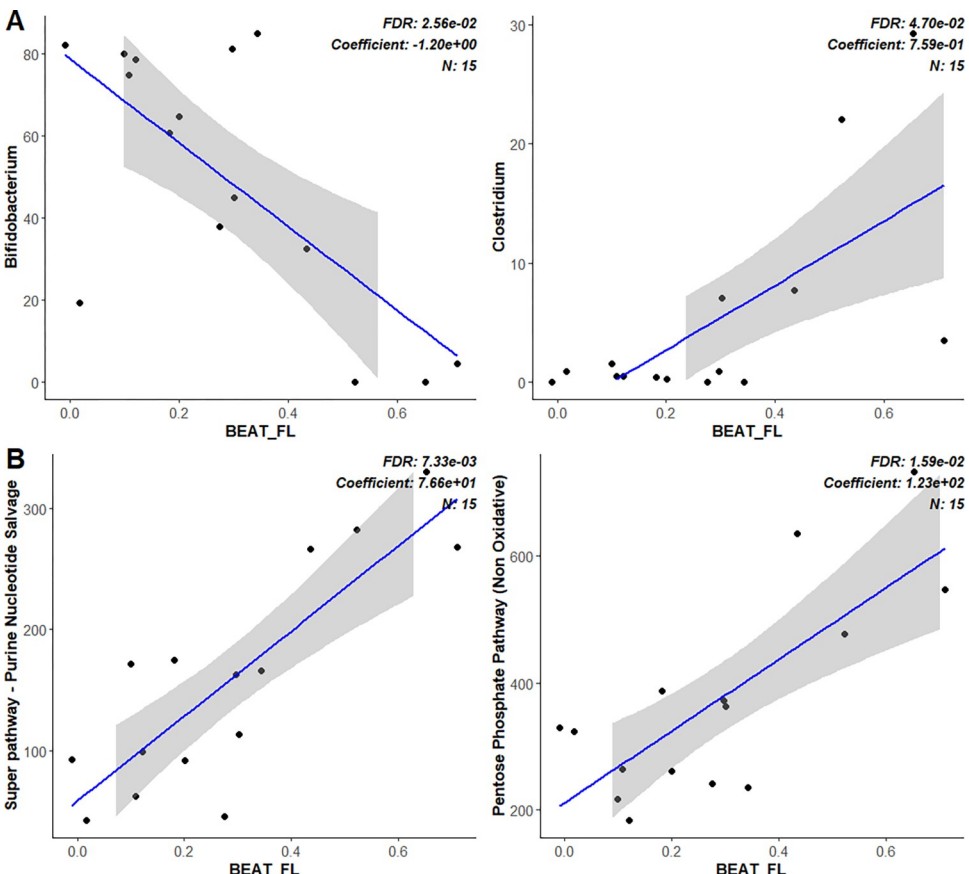

**Fig 4. Association analysis.** (A) Association analysis between bacteria abundance (CLR transformed) and EEG measurements, at the genus level. (B) Association analysis between unstratified pathways abundances and EEG measurements. Significant associations have an adjusted p-value < 0.05 and a minimum prevalence of 10%.

and Purine Nucleotide Salvage and Pentose Phosphate Pathway (Fig 4B). In the second association model, the statistical test identified 25 positive associated pathways (8 related to biosynthesis, 15 related to degradation, utilization, and assimilation, and 2 from generation of precursor metabolites and energy), and 3 negatively associated biosynthesis pathways to BEAT_FL (S2 Table). No other variables had significant associations after age correction.

Association analysis with the stratified metabolic pathways has a high number of comparisons, as each pathway is divided for each identified bacteria species, causing the statistical power of the tests to be significantly reduced. To account for the increased number of comparisons we filtered the pathways mapped to the bacteria to include those with the highest contributions to the variance of the samples, as shown in Fig 2B. For the EEG/fNIRS measurements, only the EEG measurements of BEAT_ALL, and BEAT_FL had statistically significant associations to the stratified pathways. All the associations were positive, with BEAT_ALL having 7 associations and BEAT_FL having 85. Over 40% of the associations of BEAT_FL were related to Cofactor biosynthesis and Amino Acid biosynthesis (S10 Fig). Some specific examples can be found in S8B Fig, such as the positive association between BEAT_ALL and the fatty acid beta oxidation pathway from *Klebsiella pneumoniae*, or the positive association between BEAT_FL and pyrimidine deoxyribonucleoside degradation pathway from *Klebsiella pneumoniae*. In the age corrected model, we didn't identify significant associations between the stratified pathways and any of the EEG/fNIRS variables. Both models identified similar patterns,

but the age corrected model didn't reach statistical significance after multiple testing correction. The full list of significant associations can be found in S2 Table.

## Discussion

Growing evidence has shown the importance of the gut microbiome and the vast number of metabolites it synthesizes for proper infant neurodevelopment. This preliminary and exploratory study aimed to identify associations between the early infant microbiome and emerging cognitive capacities and behaviors in a multi-disciplinary fashion. Due to the nature of the study, we caution readers not to over-interpret the results, although this pilot study identified meaningful tendencies between the microbiome and the brain systems studied, which are certainly worth exploring further.

Overall, no major differences were observed between the Point and Gaze test results and sample site (UBC, McMaster) when comparing the alpha or beta diversity (Figs 1 and 2), and no major associations were observed between success on the Point and Gaze test and the metadata collected from the infants (S1 Fig). Nevertheless, when analyzing the differential abundance between conditions we observed a slight increase in the phylum Actinobacteria, and a slight decrease of the phylum Firmicutes, in infants who showed better joint attention, although it was not statistically significant (Fig 3A). At the genus level we observed a relative increase of *Bifidobacterium* and *Eggerthella*, with a decrease in *Hungatella* and *Streptococcus* in the infants who showed better joint attention before FDR correction (S5A Fig). The increase of *Bifidobacterium* is relevant for brain development as members from this genus are known probiotics that have strong associations with host immunity and connections to the brain-gut axis [73]. Recent studies have shown the importance of *Bifidobacterium* species colonization during postnatal development as they can promote the formation of synapses and microglial function [74]. Not only was the *Bifidobacterium* genus more abundant in infants who succeed on at least one of the point and gaze subtests, but it was also among the top five bacteria that contributed to the variance between the samples (Fig 2B), with the three most commonly found species of *Bifidobacterium* in infants (*B.longum*, *B.breve* and *B.bifidum*) [74]. Higher levels of Firmicutes have been seen in infants born by C-section [75], but when comparing the samples by delivery method we didn't find any significant difference, perhaps due to our small sample size. Not much has been described regarding the genus *Eggerthella* and brain development; mostly it is described as a common anaerobe in the human intestine, with some members having pathogenic capacities [76, 77]. Lastly, the genus *Hungatella*, although statistically different between infants' success on the Point and Gaze task, is likely to be an artifact driven by the small sample population, because only seven infants had this genus and most of these were in one group (S5A Fig). It is important to note that we are using the Point and Gaze test as an indicator of brain development due to its association with later language development, but it is not a perfect measure [60]. The major benefit of analyzing this variable was the ability to compare all the infants across both sites with the same test. Although infants were tested at different facilities by different researchers, care was taken to standardize the protocol and the data was all analyzed by the UBC team, with the age of the infants being the major difference between the research sites.

For the association analyses we utilized two statistical models, the first using linear models of CLR transformed data, while the second incorporated the age of the infants as a fixed effect. For the first model there were several associations between the microbes and the EEG measurements (Fig 4A), and between the metabolic pathways and the EEG measurements (Fig 4B). From the measured variables we didn't identify any statistically significant association between the bacterial abundance or metabolic pathway abundance and measurements from

the fNIRS data. In the first statistical model, from the EEG measurements included in the association analysis, only two of the twelve had significant associations, BEAT_ALL and BEAT_FL, with the most prominent being BEAT_FL. Importantly, this variable represents the strength of the brain's representation of the basic beat frequency in the presented auditory rhythm, from which slower frequencies arising from metrical beat groupings can be constructed. Thus, we would expect the beat frequency to be present in all infants who listened to the stimulus and to be the most basic and important frequency for infants to extract from the rhythm. Larger brain amplitudes at the beat frequency indicate better or more efficient encoding of the beat, a skill that is important for tracking speech rhythms as well as musical rhythms [50, 51, 78]. Further, dyslexia and other developmental disorders are associated with poor beat tracking, and non-speech rhythmic primes can enhance language processing [31, 79–81]. For the age corrected model only BEAT_FL had significant associations with unstratified pathways; in the stratified analysis no variables had any significant associations. All the statistically significant pathways identified in the first model had a p-value under 0.05 in the age correct model before FDR correction, indicating both models identify similar patterns but didn't reach statistical significance.

At the genus level only BEAT_FL had significant associations. BEAT_FL had a strong negative association to *Bifidobacterium* abundance, and weaker associations to *Clostridium* and *Enterococcus*. The negative association to *Bifidobacterium* was unexpected, as this genus is relevant for brain development, host immunity, etc., and we would expect that higher values of BEAT_FL to be a positive marker for brain development [73, 74, 82]. At the species level the power of the analysis is low, as not all the samples share the same bacteria. BEAT_FL positively associated to *Enterococcus faecalis*, a natural occurring intestinal bacteria that can become a pathobiont under dysbiosis [83]; nevertheless strains of *E.faecalis* have been considered potential candidate probiotics for their ability to produce bacteriocins that can prevent the growth of other pathogenic organisms [84].

From the unstratified pathways only BEAT_FL had statistically significant associations. Specifically, BEAT_FL had positive associations with pathways related to biosynthesis, degradation/assimilation and generation of precursor metabolites and energy production, while only having negative associations to pathways related to biosynthesis (S9 Fig). When analyzing the top associations based on their effect size (model coefficient value) [67] we observed pathways such as the increase of genes related to the pentose phosphate pathways. This is an important pathway of glucose metabolism in the brain that produces other metabolites required for the synthesis of new nucleotides or the generation of antioxidative agents [85]. In the gut, this pathway is key for the metabolism of dietary fibers, which are broken down into hexoses and pentoses [26, 27, 86, 87]. From these compounds the bacteria then produce important metabolites, such as SCFAs (metabolites involved in the microbiota-gut-brain axis due to neuroactive properties), that can interact with structures like the blood brain barrier that controls the movement of nutrients/molecules to the brain and is directly linked to proper brain development [26, 27, 86, 87]. The other top associations with BEAT_FL were pathways related to nucleosides/nucleotides degradation or salvage (S2 Table). Nucleosides/nucleotides are essential nutrients during times of rapid growth and development, such as during infants' early life [88–90]. The increase of these pathways may be explained as an increase in the nucleotide/nucleoside metabolism to degrade the metabolites from the food source and salvage it for own usage, instead of *de novo* synthesis of the compound.

In the complete stratified pathway analysis, we didn't identify any significant association with the EEG or fNIRS data, likely due to the increased number of comparisons the model has to correct for, as all the counts for each analyzed pathway are divided between all the bacteria that contribute to that metabolic pathway.

When studying the stratified pathways for the top contributing bacteria we found associations with the variables BEAT_ALL and BEAT_FL. Both BEAT_ALL and BEAT_FL had associations with pathways associated only to *Klebsiella pneumoniae*. As before, we saw similar pathways associating, such as the pentose phosphate pathways, branched chain amino acid biosynthesis, and the nucleotide/nucleoside degradation/salvage (S2 Table), as well as other important pathways such as fatty acid beta oxidation and cofactor and vitamin biosynthesis. Fatty acid beta oxidation is relevant at a young age as newborns have a change of diet when introduced to milk, as its composition is high on fats and low on carbohydrates [91]. The increase of this metabolic pathway can indicate a rise in ketone production by the microbiome, which can be helpful as a supply of energy, especially for the brain during its continuing development [91]. Cofactor and vitamin biosynthesis upregulation has been described in infants with high predictive cognitive performance based on the MSEL, as metabolites such as folate are important for neurodevelopment [28, 92].

The identified associations between the microbiome and infant brain development must be interpreted with caution, as low sample size and unmeasured variables could have a significant effect on the results. The infant microbiome is a dynamic environment that can be modified by external components, such as the diet of the mothers during pregnancy and lactation, the type of infant-feeding and even the interactions they have with others and the spaces they live.

Infant diet is a crucial factor that not only changes the microbiome but is important for infant development and health. This is seen in the beneficial effects of breastfeeding in short- and long-term health, in providing both commensal microbiomes and prebiotic components such as oligosaccharides that help stablish a healthy microbiome, as well as enhanced neurodevelopment [93–95]. When evaluating the effect of infant feeding (breast milk versus mix feeding, and the introduction to solid foods) in the association models for the bacteria and pathway analysis we didn't observe significative associations after FDR correction for any of the measurements. This was driven by the small sample size for the previously significant variables (BEAT_FL and BEAT_ALL), as the top hits before FDR correction were highly maintained between models (over 80% overlap for the pathway analyses), with metabolic pathways of L-lysine biosynthesis having increased effect after feeding correction for the BEAT_FL variable.

Maternal diet during pregnancy and lactation have been shown to influence the infant microbiome and neurodevelopment [93, 96–99]. Diet can modulate the gut microbiome of the mothers, which in turn can modify the microbiome of the human milk via the entero-mammary pathway and add beneficial metabolites derived from the gut such as SCFA [96, 97]. During pregnancy maternal diet is crucial for the proper development of the infants, as their systems, especially the brain, are highly susceptible to nutritional alterations [93, 98, 99]. It has also been observed that high quality diets are associated with improvements to certain functions such as visual spatial skills [99]. Although not measured in this study, maternal diet could be a confounding factor in the identified associations and an interesting factor to consider in future analysis.

In summary, we found no differences in the alpha or beta diversity when comparing infants who were successful or not on the Point and Gaze task, a behavioral test of joint attention in a social situation. However, when comparing the microbial composition, we found an increase of *Bifidobacterium* and *Eggerthella* in infants who succeeded on at least one of the point and gaze subtests, while having a lower abundance of *Hungatella* and *Streptococcus*. In the association analysis we found several pathways that can be linked to brain development, either because of their fundamental role as building blocks for growth (in the case of the essential amino acids) or by their impact in the synthesis of metabolites that influence brain development such as SCFAs (by the pentose phosphate pathway) or branched chain amino acids. We

did not find any associations between the metagenomic measures and the fNIRS measures of language processing in the infant brain. However, across a number of analyses we found one predominant EEG measure that consistently associated to the metagenomic data, the amplitude of the EEG at the beat frequency of the presented rhythmic stimulus in frontal left sites (BEAT_FL). BEAT_FL can be interpreted as a metric of brain development because neural tracking of rhythm is critical for learning both music and speech and poor rhythmic processing is associated with most major developmental disorders. Further, the left lateralization of the neural beat tracking associations is consistent with the left lateralization of neural tracking reported in Flaten *et. al.* (2022), where the amplitudes of brain responses to rhythms were larger in the left hemisphere [40]. Perhaps of most interest in the results are the associations between BEAT_FL and pathways related to bacterial pentose phosphate pathways, which lead to the formation of metabolites with neuroactive properties that can modulate the brain, such as SCFA, and the nucleosides/nucleotides salvage, which is necessary to maintain the levels of nucleotides in times or rapid growth and development, among other pathways [26, 27, 88–90].

The results need to be treated with caution due to the relatively small sample size. Nevertheless, many interesting associations were identified between the microbiome and brain function in early infancy, indicating that replication and further research could be fruitful for understanding the role of the microbiome in early cognitive development.

## Supporting information

**S1 Fig. Odds ratio analysis.** Odds ratio analysis between several metadata variables (collection site, children with older siblings, having pets, children introduced to solid foods, sex (FEM), children mode of delivery and age) and the success on the gaze (A) or point (B) subtest. Only those related to delivery type were significant.
(TIF)

**S2 Fig. Taxonomic richness analysis.** Taxonomic richness analysis comparing those who were unsuccessful versus successful on the point subtest (A), the gaze subtest (B), at least one of the point and gaze subtests (C), as well as comparing the collection sites (D). A two-tailed t-test was performed between the successful and failed attempts. Taxonomic richness had a p-value < 0.05 between collection sites, there was not a statistical difference between the other groups.
(TIF)

**S3 Fig. Simpson index analysis.** Microbial diversity analysis using Simpson index comparing those who were unsuccessful versus successful on the point subtest (A), the gaze subtest (B), at least one of the point and gaze subtests (C), as well as comparing the collection sites (D). A two-tailed t-test was performed between the successful and failed attempts. There was not a statistical difference between the groups.
(TIF)

**S4 Fig. Shannon index analysis.** Functional diversity analysis using Shannon index comparing those who were unsuccessful versus successful on the point subtest (A), the gaze subtest (B), at least one of the point and gaze subtests (C), as well as comparing the collection sites (D). A two-tailed t-test was performed between the successful and failed attempts. There was not a statistical difference between the groups.
(TIF)

**S5 Fig. Bacteria relative abundance.** Bacteria relative abundance at the genus level of the most abundant microorganisms in the samples. Difference in bacteria relative abundance by

success on at least one of the point and gaze subtests.
(TIF)

**S6 Fig. EEG and fNIRS association analysis.** Association analysis between bacteria abundance (CLR transformed) and EEG/fNIRS measurements at the genus level.
(TIF)

**S7 Fig. EEG and fNIRS association analysis.** Association analysis between bacteria abundance (CLR transformed) and EEG/fNIRS measurements at the species level.
(TIF)

**S8 Fig. Association analysis.** (A) Association analysis between bacteria abundance (CLR transformed) and EEG measurements at the species level. (B) Association analysis between stratified pathways abundances and EEG measurements. Significant associations have an adjusted p-value < 0.05 and a minimum prevalence of 10%.
(TIF)

**S9 Fig. Association analysis.** Association analysis between unstratified pathway abundances and EEG measurements. A) Associated pathways at the Super Class level and the variable BEAT_FL faceted by negative or positive associations. B) Association at the metabolism level. C) Association at the Class level.
(TIF)

**S10 Fig. Association analysis.** Association analysis between stratified pathway abundances and EEG. The analyzed pathways come from the top five contributing bacteria, found in Fig 4. Association at the Class level for the variables BEAT_ALL and BEAT_FL.
(TIF)

**S1 Table. EEG measurements legend.**
(CSV)

**S2 Table. Microbial and pathway associations to EEG measurements.**
(XLSX)

## Acknowledgments

We thank Nicole Sugden for her leadership role in collecting the fNIRS data. We also thank the families for their participation in this study. B.B.F. is a CIFAR Senior Fellow and a University of British Columbia Peter Wall Distinguished Professor. J.F.W is on the advisory committee of the Brain Mind and Consciousness CIFAR program and a University of British Columbia Killam Professor. L.J.T. is a CIFAR Senior Fellow and a McMaster Distinguished University Fellow. S.H. is supported by the International Tuition Award at UBC. E.F. was supported by a Canadian graduate scholarship from SSHRC, an Ontario Graduate Scholarship, as well as an NSERC CREATE award in Complex Dynamics.

## Author Contributions

**Conceptualization:** Charisse Petersen.

**Data curation:** Charisse Petersen.

**Formal analysis:** Sebastian Hunter, Erica Flaten, Charisse Petersen, Judit Gervain.

**Funding acquisition:** Janet F. Werker, Laurel J. Trainor, Brett B. Finlay.

**Investigation:** Erica Flaten, Charisse Petersen.

**Methodology:** Charisse Petersen, Judit Gervain.

**Supervision:** Janet F. Werker, Laurel J. Trainor, Brett B. Finlay.

**Visualization:** Sebastian Hunter.

**Writing – original draft:** Sebastian Hunter.

**Writing – review & editing:** Erica Flaten, Judit Gervain, Janet F. Werker, Laurel J. Trainor, Brett B. Finlay.

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
