## [Decision Letter · Decision Letter 0]

29 May 2023

PONE-D-23-13858Babies, Bugs and Brains: How the early microbiome influences infant brain and behavior developmentPLOS ONE

Dear Dr. Hunter,

Thank you for submitting your manuscript to PLOS ONE. After careful consideration, we feel that it has merit but does not fully meet PLOS ONE’s publication criteria as it currently stands. Therefore, we invite you to submit a revised version of the manuscript that addresses the points raised during the review process.

We look forward to receiving your revised manuscript.

Kind regards,

Brenda A Wilson, Ph.D.

Academic Editor

PLOS ONE

Journal Requirements:

   "We thank Nicole Sugden for her leadership role in collecting the fNIRS data. We also thank the families for their participation in this study. This study was supported by Canadian Institute for Advance Research (CIFAR) Grants. Work in B.B.F.’s lab is also supported by a Canadian Institute for Health Research (CIHR) Foundation Grant, and in J.F.W.’s lab by a NSERC DG. B.B.F. is a CIFAR Senior Fellow and a University of British Columbia Peter Wall Distinguished Professor. S.H. is supported by the International Tuition Award at UBC. Work in L.J.T.’s lab was supported by grants from the Canadian Institutes of health Research (CIHR), the Natural Sciences and Engineering Research Council of Canada (NSERC) and the Social Sciences and Humanities Research Council of Canada (SSHRC). L.J.T. is a CIFAR Fellow and a McMaster Distinguished University Fellow. E.F. was supported by a Canadian graduate scholarship from SSHRC, an Ontario Graduate Scholarship, as well as an NSERC CREATE award in Complex Dynamics."

   "This research was funded by grants to JFW, LJT and BBF from the Canadian Institute for Advanced Research (CIFAR, https://cifar.ca/) (FL-000981-CF, FL-000982-CF, & FL-000983-CF). The funders had no role in study design, data collection and analysis, decision to publish, or preparation of the manuscript."

Additional Editor Comments:

Both reviewers noted a number of issues that if adequately addressed would significantly improve the manuscript. Please revise the manuscript accordingly.

Reviewers' comments:

Reviewer's Responses to Questions

**Comments to the Author**

1. Is the manuscript technically sound, and do the data support the conclusions?

Reviewer #1: Yes

Reviewer #2: Yes

2. Has the statistical analysis been performed appropriately and rigorously? 

Reviewer #1: Yes

Reviewer #2: Yes

3. Have the authors made all data underlying the findings in their manuscript fully available?

Reviewer #1: Yes

Reviewer #2: Yes

4. Is the manuscript presented in an intelligible fashion and written in standard English?

Reviewer #1: Yes

Reviewer #2: Yes

5. Review Comments to the Author

Reviewer #1: Hunter S. and colleagues submitted an original manuscript describing how the microbiome in early infancy (around 4-5 months of age on average in human babies) influences the brain development and behaviour. They collected useful data from a total of 40 babies in two different centres. From their preliminary exploratory study they honestly suggest potential important aspects of the gut-microbiome-brain interactions that could impact the infant brain development and behaviours in early and potentially also later life stages. They in fact have highlighted some characteristics of the microbiota composition and functions that could be important in this phenomenon.

I have couple of questions:

1. the samples have been collected and the babies have been screened during the pandemic (the recruitment started late 2019 and finished in October 2020): could the authors comment on perhaps some effects of the pandemics on both microbiota composition (heavy usage of masks and disinfectants, day cares were closed -most likely also in Canada- so babies were exposed less to other babies/individuals outside their own family and infections in day care settings) and perhaps also on the brain development of these babies? do similar data exist on other cohorts collected perhaps not during the pandemic that could be useful to compare?

2. are data on the maternal diet available that could be cofounding effect of the babies microbiota/brain development and behaviour? if not, could the authors comment this point in the discussion?

3. maybe I missed it, but could the data being correlated also on the only breast milk feeding or mixed feeding or only solid food feeding? It was clear to me that data on the solid food introduction were available but it is not so clear to me at which extend the babies could be classified in the 3 groups I have mentioned here, if it is possible at all. If not, could the authors comment on the importance of diet in this early life stage and on the microbiota and brain development and behaviours.

The manuscript is very well written and easy to follow and it would be interesting to expand such a study in a bigger one and collect data from the same babies also later on in life to follow the trajectory of their microbiota and brain development.

Reviewer #2: The authors aimed to investigate associations between infant microbiome and early cognitive development. The authors used three measures of early cognitive development (Point and Gaze, fNIRs, and EEG) and tested its association with gut microbiome abundance and diversity, measured by sequencing of the fecal microbiome. Although the sample size is small, the authors identified some interesting associations that would be of great interest for further studies. For example, Point and Gaze test success in infants is associated with increased Actinobacteria and reduced Firmicutes. The study is well-designed and analyzed, and the authors are careful when drawing conclusions. I would recommend its publication after the below comments are addressed.

Comments

1. The title alludes to a causative influence of microbes on early brain development. I would suggest changing it to more accurately describe the data and conclusion of the current study.

2. References would be helpful for the statements on line 484 - “to Bifidobacterium was unexpected, as this genus is relevant for brain development, host immunity etc.”

3. Line 540-543. The pentose phosphate pathway identified from the pathway analysis belongs to the gut microbes, and thus should not be directly involved in brain glucose metabolism. The authors should clarify how microbial pentose phosphate pathway relate to brain.

4. Minor comment: The order of findings in the result/conclusion section is different from that described in the abstract. Maybe rearrange in abstract.

6. PLOS authors have the option to publish the peer review history of their article (what does this mean?). If published, this will include your full peer review and any attached files.

Reviewer #1: **Yes: **Francesca Ronchi

Reviewer #2: No

---

## [Author Response · Author response to Decision Letter 0]

29 Jun 2023

The following answer is the same found in the 'Response to Reviewers' file submitted in the attached files:

PLOS ONE

Editorial Office

29 June 2023

Dear Editor,

We are pleased to resubmit our revised manuscript PONE-D-23-13858 entitled “Babies, Bugs and Brains: How the early microbiome associates with infant brain and behavior development” originally submitted May 6, 2023. 

We are grateful for the excellent, rapid, and thorough critique provided by the Reviewers. This subsequent revised version has addressed their queries, including possible confounding effects, corrected the sample collection dates and added and corrected the bibliography. We have now addressed all the comments in the revised version which we feel has improved the manuscript. Please find attached a point-by-point response to how we addressed all their comments.

Once again, we thank the Reviewers for their thoughtful critique, editorial suggestions, as well as supportive comments. Thank you for your consideration.

Yours sincerely,

Sebastian Hunter

 

Reviewers Comments:

Reviewer #1

Hunter S. and colleagues submitted an original manuscript describing how the microbiome in early infancy (around 4-5 months of age on average in human babies) influences the brain development and behaviour. They collected useful data from a total of 40 babies in two different centres. From their preliminary exploratory study they honestly suggest potential important aspects of the gut-microbiome-brain interactions that could impact the infant brain development and behaviours in early and potentially also later life stages. They in fact have highlighted some characteristics of the microbiota composition and functions that could be important in this phenomenon.

- We thank the Reviewer for this kind comment.

I have couple of questions:

1. The samples have been collected and the babies have been screened during the pandemic (the recruitment started late 2019 and finished in October 2020): could the authors comment on perhaps some effects of the pandemics on both microbiota composition (heavy usage of masks and disinfectants, day cares were closed -most likely also in Canada- so babies were exposed less to other babies/individuals outside their own family and infections in day care settings) and perhaps also on the brain development of these babies? do similar data exist on other cohorts collected perhaps not during the pandemic that could be useful to compare?

- We thank the Reviewer for this comment. The dates incorporated in the original manuscript regarding the data collection from the fecal samples were unfortunately incorrect (thanks for catching this). The correct dates were between February 15, 2019, and February 24, 2020. All data was collected prior to the pandemic (March 11, 2020, COVID-19 was declared a global pandemic) (Lines 136, 161-163, and 192-195).

2. Are data on the maternal diet available that could be cofounding effect of the babies microbiota/brain development and behaviour? if not, could the authors comment this point in the discussion?

- We thank the Reviewer for this comment. Regarding maternal diet we didn’t collect data, but we explored the possible confounding effects it could have in our results, as the mother’s diet can influence both the infant’s microbiome and its development. It was identified as an interesting factor to include in future analysis (addressed on Line 542-550).

3. Maybe I missed it, but could the data being correlated also on the only breast milk feeding or mixed feeding or only solid food feeding? It was clear to me that data on the solid food introduction were available but it is not so clear to me at which extend the babies could be classified in the 3 groups I have mentioned here, if it is possible at all. If not, could the authors comment on the importance of diet in this early life stage and on the microbiota and brain development and behaviours. 

- We thank the Reviewer for this comment. Regarding infants feeding and introduction to solid foods, we included these variables in another statistical model used for the association analysis and didn’t identify any significant difference in any of the variables after FDR correction. Nevertheless, when we compared the top hits from the models before FDR correction, they were highly similar (with over 80% overlap) for the BEAT_FL and BEAT_ALL variables (the only variables with statistically significant variables in the first model). This indicates that due to small sample sizes the statistical test didn’t have enough power after FDR correction, but the tendencies were mostly the same. This observation has been added to the discussion (addressed on Line 532-541). 

The manuscript is very well written and easy to follow and it would be interesting to expand such a study in a bigger one and collect data from the same babies also later on in life to follow the trajectory of their microbiota and brain development.

- We thank the Reviewer for this kind comment.

Reviewer #2

The authors aimed to investigate associations between infant microbiome and early cognitive development. The authors used three measures of early cognitive development (Point and Gaze, fNIRs, and EEG) and tested its association with gut microbiome abundance and diversity, measured by sequencing of the fecal microbiome. Although the sample size is small, the authors identified some interesting associations that would be of great interest for further studies. For example, Point and Gaze test success in infants is associated with increased Actinobacteria and reduced Firmicutes. The study is well-designed and analyzed, and the authors are careful when drawing conclusions.

- We thank the Reviewer for this kind comment.

I would recommend its publication after the below comments are addressed.

1. The title alludes to a causative influence of microbes on early brain development. I would suggest changing it to more accurately describe the data and conclusion of the current study.

- We thank the Reviewer for this comment. We changed the title to ‘Babies, Bugs and Brains: How the early microbiome associates with infant brain and behavior development', to not allude to a causative influence of the microbiome, but to the observed associations between the microbiome and brain development (Line 1-2).

2. References would be helpful for the statements on line 484 - “to Bifidobacterium was unexpected, as this genus is relevant for brain development, host immunity etc.”

- We thank the Reviewer for this comment. We added references as suggested, to reinforce the beneficial properties this genus has for brain development and host immunity (Line 486-489).

3. Line 540-543. The pentose phosphate pathway identified from the pathway analysis belongs to the gut microbes, and thus should not be directly involved in brain glucose metabolism. The authors should clarify how microbial pentose phosphate pathway relate to brain.

- We thank the Reviewer for this comment. We removed the previous statement of the importance of pentose phosphate pathway that was related to the brain glucose metabolism and replaced it with the microbial importance. In the gut this pathway is important due to the metabolism of pentose sugars from diet fibers, which end as short chain fatty acids (SCFA), metabolites with neuroactive active properties that can modulate the brain and the blood brain barrier (Line 567-571). 

4. Minor comment: The order of findings in the result/conclusion section is different from that described in the abstract. Maybe rearrange in abstract.

- We thank the Reviewer for this comment. The abstract was rearranged to follow the same order as the one presented in the results.

Journal Requirements

- We thank the Academic Editor for this comment. Regarding PLOS ONE’s style requirements we ensured all the files’ naming was correct according to the templates provided above. We also ensured the manuscript has the correct format for all its sections.

 "We thank Nicole Sugden for her leadership role in collecting the fNIRS data. We also thank the families for their participation in this study. This study was supported by Canadian Institute for Advance Research (CIFAR) Grants. Work in B.B.F.’s lab is also supported by a Canadian Institute for Health Research (CIHR) Foundation Grant, and in J.F.W.’s lab by a NSERC DG. B.B.F. is a CIFAR Senior Fellow and a University of British Columbia Peter Wall Distinguished Professor. S.H. is supported by the International Tuition Award at UBC. Work in L.J.T.’s lab was supported by grants from the Canadian Institutes of health Research (CIHR), the Natural Sciences and Engineering Research Council of Canada (NSERC) and the Social Sciences and Humanities Research Council of Canada (SSHRC). L.J.T. is a CIFAR Fellow and a McMaster Distinguished University Fellow. E.F. was supported by a Canadian graduate scholarship from SSHRC, an Ontario Graduate Scholarship, as well as an NSERC CREATE award in Complex Dynamics."

 "This research was funded by grants to JFW, LJT and BBF from the Canadian Institute for Advanced Research (CIFAR, https://cifar.ca/) (FL-000981-CF, FL-000982-CF, & FL-000983-CF). The funders had no role in study design, data collection and analysis, decision to publish, or preparation of the manuscript."

- We thank the Academic Editor for this comment. We would like to change the Acknowledgement and Funding section, as there was funding information in the Acknowledgement section that should be in the Funding section. The current version of the Acknowledgement and Funding sections are as follow:

o Acknowledgements: We thank Nicole Sugden for her leadership role in collecting the fNIRS data. We also thank the families for their participation in this study. B.B.F. is a CIFAR Senior Fellow and a University of British Columbia Peter Wall Distinguished Professor. J.F.W is on the advisory committee of the Brain Mind and Consciousness CIFAR program and a University of British Columbia Killam Professor. L.J.T. is a CIFAR Senior Fellow and a McMaster Distinguished University Fellow. S.H. is supported by the International Tuition Award at UBC. E.F. was supported by a Canadian graduate scholarship from SSHRC, an Ontario Graduate Scholarship, as well as an NSERC CREATE award in Complex Dynamics.

o Funding Statement: This research was funded by grants to JFW, LJT and BBF from the Canadian Institute for Advanced Research (CIFAR, https://cifar.ca/) (FL-000981-CF, FL-000982-CF, & FL-000983-CF). Work in B.B.F.’s lab is also supported by a Canadian Institute for Health Research (CIHR) Foundation Grant, in J.F.W.’s lab by a Natural Sciences and Engineering Research Council of Canada (NSERC) Discovery Grant and a Social Sciences and Humanities Research Council of Canada (SSHRC) Insight Grant, and in L.J.T.’s lab by grants from CIHR, NSERC and SSHRC. The funders had no role in study design, data collection and analysis, decision to publish, or preparation of the manuscript.

- We thank the Academic Editor for this comment. The repository information of our data will be provided at acceptance of the manuscript.

- We thank the Academic Editor for this comment. The ethics statement was added as an item in the Methods of the manuscript with the full name of the ethics committee and the identification number (Line 148-151).

- We thank the Academic Editor for this comment. The reference list was updated with new entries that were added in the extended discussion section in the correct style (Vancouver). The reference list was modified to follow PLOS guidelines, including only the first six authors followed by et al. The added or modified references are the following:

o Added:

Stuivenberg GA, Burton JP, Bron PA, Reid G. Why are bifidobacteria important for infants?. Microorganisms. 2022 Jan 25;10(2):278.

Belfort MB. The science of breastfeeding and brain development. Breastfeeding Medicine. 2017 Oct 1;12(8):459-61.

O'Sullivan A, Farver M, Smilowitz JT. Article Commentary: The influence of early infant-feeding practices on the intestinal microbiome and body composition in infants. Nutrition and metabolic insights. 2015 Jan;8:NMI-S29530.

Van den Elsen LW, Garssen J, Burcelin R, Verhasselt V. Shaping the gut microbiota by breastfeeding: the gateway to allergy prevention?. Frontiers in pediatrics. 2019:47.

Lundgren SN, Madan JC, Emond JA, Morrison HG, Christensen BC, Karagas MR, et al. Maternal diet during pregnancy is related with the infant stool microbiome in a delivery mode-dependent manner. Microbiome. 2018 Dec;6:1-1.

Sindi AS, Geddes DT, Wlodek ME, Muhlhausler BS, Payne MS, Stinson LF. Can we modulate the breastfed infant gut microbiota through maternal diet?. FEMS Microbiology Reviews. 2021 Sep;45(5):fuab011.

Arija V, Canals J. Effect of Maternal Nutrition on Cognitive Function of Children. Nutrients. 2021 May 13;13(5):1644.

Mahmassani HA, Switkowski KM, Scott TM, Johnson EJ, Rifas-Shiman SL, Oken E, et al. Maternal diet quality during pregnancy and child cognition and behavior in a US cohort. The American Journal of Clinical Nutrition. 2022 Jan;115(1):128-41.

o Modified:

Mullen EM. Mullen scales of early learning. Circle Pines, MN: AGS; 1995. (Previously: Shank L. Mullen scales of early learning. Encyclopedia of clinical neuropsychology. 2011:1669-71.)

Comeau AM, Douglas GM, Langille MG. Microbiome helper: a custom and streamlined workflow for microbiome research. MSystems. 2017 Feb 28;2(1):e00127-16. (Previously: LangilleLab. (2020). Metagenomics standard operating procedure V2 · LangilleLab. GitHub. Retrieved from https://github.com/LangilleLab/microbiome_helper/wiki/Metagenomics-standard-operating-procedure-v2)

McIver LJ, Abu-Ali G, Franzosa EA, Schwager R, Morgan XC, Waldron L, et al. bioBakery: a meta’omic analysis environment. Bioinformatics. 2018 Apr 1;34(7):1235-7. (Previously: bioBakery. (2022). Welcome to the HUMAnN 3.0 tutorial· bioBakey. GitHub. Retrieved from https://github.com/biobakery/biobakery/wiki/humann3)

R Core Team. R: A language and environment for statistical computing. R Foundation for Statistical Computing, Vienna, Austria. http://www. R-project. org/. 2016. (Previously: R Core Team (2022). R: A language and environment for statistical computing. R Foundation for Statistical Computing, Vienna, Austria. URL https://www.R-project.org/.)

Oksanen J. Vegan: community ecology package. http://CRAN. R-project. org/package= vegan. 2010. (Previously: Oksanen J., et al. (2022). vegan: Community Ecology Package. R package version 2.6-2, https://CRAN.R-project.org/package=vegan)

Kassambara A. Factoextra: extract and visualize the results of multivariate data analyses. R package version. 2016;1. (Previously: Kassambara A, Mundt F (2020). factoextra: Extract and Visualize the Results of Multivariate Data Analyses. R package version 1.0.7, https://CRAN.R-project.org/package=factoextra)

Wickham H, Chang W, Wickham MH. Package ‘ggplot2’. Create elegant data visualisations using the grammar of graphics. Version. 2016;2(1):1-89. (Previously: H. Wickham. ggplot2: Elegant Graphics for Data Analysis. Springer-Verlag New York, 2016.)

Mallick H, Rahnavard A, McIver LJ, Ma S, Zhang Y, Nguyen LH, et al. Multivariable association discovery in population-scale meta-omics studies. PLoS computational biology. 2021 Nov 16;17(11):e1009442. (Removed duplicate, was previously: Mallick H, Rahnavard A, McIver LJ (2020). MaAsLin 2: Multivariable Association in Population-scale Meta-omics Studies. R/Bioconductor package, http://huttenhower.sph.harvard.edu/maaslin2 and Mallick H, Rahnavard A, McIver LJ, Ma S, Zhang Y, Nguyen LH, Tickle TL, Weingart G, Ren B, Schwager EH, Chatterjee S. Multivariable association discovery in population-scale meta-omics studies. PLoS computational biology. 2021 Nov 16;17(11):e1009442.)

---

## [Editor Report · Decision Letter 1]

2 Jul 2023

Babies, Bugs and Brains: How the early microbiome associates with infant brain and behavior development

PONE-D-23-13858R1

Dear Dr. Hunter,

We’re pleased to inform you that your manuscript has been judged scientifically suitable for publication and will be formally accepted for publication once it meets all outstanding technical requirements.

Kind regards,

Brenda A Wilson, Ph.D.

Academic Editor

PLOS ONE

Additional Editor Comments (optional):

It appears that all previous reviewer concerns have been adequately addressed.

---

## [Editor Report · Acceptance letter]

17 Jul 2023

PONE-D-23-13858R1 

Babies, Bugs and Brains: How the early microbiome associates with infant brain and behavior development 

Dear Dr. Hunter:

I'm pleased to inform you that your manuscript has been deemed suitable for publication in PLOS ONE. Congratulations! Your manuscript is now with our production department. 

Kind regards, 

on behalf of

Dr. Brenda A Wilson 

Academic Editor

PLOS ONE